# REINFORCEMENT LEARNING VIA IMPLICIT IMITATION GUIDANCE

## ABSTRACT

We study the problem of sample efficient reinforcement learning, where prior data such as demonstrations are provided for initialization in lieu of a dense reward signal. A natural approach is to incorporate an imitation learning objective, either as regularization during training or to acquire a reference policy. However, imitation learning objectives can ultimately degrade long-term performance, as it does not directly align with reward maximization. In this work, we propose to use prior data solely for guiding exploration via noise added to the policy, sidestepping the need for explicit behavior cloning constraints. The key insight in our framework, Data-Guided Noise (DGN), is that demonstrations are most useful for identifying which actions should be explored, rather than forcing the policy to take certain actions. Our approach achieves up to 2-3x improvement over prior reinforcement learning from offline data methods across seven simulated continuous control tasks.

## 1 INTRODUCTION

Progress in deep reinforcement learning (RL) has led to considerable success in a wide range of complex domains from games such as Go (Silver et al., 2016) to large-scale language model alignment (Ouyang et al., 2022). However, applying RL to real-world continuous control tasks remains difficult due to poor sample efficiency and the challenge of sparse rewards. One attractive framework for addressing these issues is to leverage information from an offline dataset consisting of previously collected data such as expert demonstrations.

Existing methods that leverage demonstrations in online RL often either underuse or overconstrain with them. A widely used approach is to initialize the replay buffer with demonstration data and oversample from it during off-policy training (Vecerik et al., 2017; Nair et al., 2018; Ball et al., 2023). While this can provide some early guidance, it only indirectly leverages the information in the demonstrations. On the other hand, imitation learning (IL) regularized methods use demonstrations directly, adding behavior cloning losses or regularization that constrain the policy to remain close to the expert distribution (Hester et al., 2018; Nair et al., 2020). Though these constraints can accelerate early learning, they often degrade long-term performance as the constraints do not directly align with reward maximization. More recent approaches train a separate IL reference policy to guide RL exploration (Zhang et al., 2023; Hu et al., 2023), but these require training strong IL policies and a reliable way of choosing between the IL and RL policies.

In this work, we propose Data-Guided Noise (DGN), a framework for leveraging the prior data to guide RL exploration with implicit imitation signals. Our key insight is that prior data such as expert demonstrations are especially valuable for revealing which exploratory actions are likely to be effective, particularly in sparse-reward environments–not necessarily for prescribing the final optimal behavior. Rather than imitating demonstration actions directly or regularizing the policy to stay close to the demonstration distribution, we focus on the difference between demonstration actions and the agent's current policy actions at demonstrated states. These differences can be interpreted as directions in action space that have led to successful outcomes. As a practical instantiation of DGN, we train a state-dependent covariance matrix on dataset-policy action differences and use it to inject structured noise into agent actions during rollouts. We can view this instantiation as *learning the mean of the policy via RL and the variance via imitation for that mean.* This allows the agent to follow its own learned behavior while increasing the probability of the policy taking actions that are represented in the demonstration data, improving the chance of finding a high-reward action

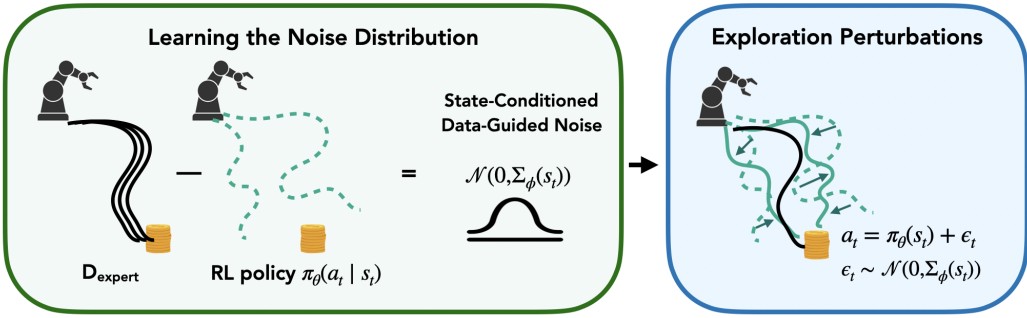

Figure 1: **Data-Guided Noise (DGN)**. We propose to guide exploration by learning a state-conditioned noise distribution that uses the difference between expert actions and the current RL policy to provide implicit imitation signals for exploration.

distribution. Crucially, by shaping the agent's exploration behavior rather than its policy optimization, DGN avoids common pitfalls of imitation-augmented RL: it does not require a strong IL policy and mechanism for switching between policies and avoids regularization that does not align with reward maximization.

Our main contribution is a framework for using implicit imitation signals to guide an RL agent towards demonstration-like regions through state-conditioned noise. Instead of enforcing imitation via a loss, we propose using differences between RL actions and expert actions to guide exploration of online RL. We evaluate DGN on several sparse-reward continuous control tasks, known to be difficult for standard RL algorithms. DGN is complementary to prior pipelines and can be integrated into standard online RL or on top of imitation-augmented RL approaches. Through evaluating on challenging benchmarks, we find that DGN matches or outperforms existing state-of-the-art approaches that use demonstration data, providing up to 2-3x improvement in performance.

## 2 RELATED WORKS

**Exploration and Sampling in Reinforcement Learning.** Effective exploration is a longstanding challenge in RL, especially in environments with sparse rewards. One line of work uses count-based bonuses to encourage visitation to novel states (Bellemare et al., 2016; Tang et al., 2017; Burda et al., 2018; Ecoffet et al., 2019), or intrinsic rewards such as the learning progress of the agent (Lopes et al., 2012; Oudeyer, 2018), model uncertainty (Schmidhuber, 2010; Houthooft et al., 2016; Pathak et al., 2019; Sekar et al., 2020), information gain (Houthooft et al., 2016), auxiliary tasks (Riedmiller et al., 2018), generating and reaching goals (Pong et al., 2020; Chen et al., 2020), and state distribution matching (Lee et al., 2019). While these methods provide general-purpose mechanisms for incentivizing exploration, they often overlook structure that may be available in the form of expert demonstrations or prior task knowledge. There are also more structured exploration approaches that guide exploration via learning from prior task structure (Vezzani et al., 2019; Singh et al., 2020). In this work, we shape the agent's action distribution towards expert-like behaviors, enabling more targeted, reward-relevant exploration.

**Offline-to-Online Reinforcement Learning.** When rewards are sparse, the aforementioned exploration strategies may struggle to efficiently discover task solutions. One remedy is to leverage prior data, such as expert demonstrations or offline collected experience, to bootstrap the learning process. Such prior data can help the agent more quickly focus on relevant states, and has been theoretically shown to improve sample efficiency (Song et al., 2022). However, naively including offline data in online RL pipelines can be unstable, leading to recent research on better approaches for offline-to-online RL. Several approaches maintain separate exploration and agent policies to balance optimism and pessimism between the two (Yang et al., 2023; Mark et al., 2023) or calibrate value estimates learned using the offline data (Nakamoto et al., 2023). AWAC and similar methods (Nair et al., 2020) find that one effective approach is to constrain policy updates using the advantage of offline actions. Lee et al. (2022) use a balanced replay and pessimistic Q-ensemble to address the state-action distribution shift when fine-tuning online. Another common recipe is to initialize the

replay buffer with demonstrations and oversample them during off-policy training (Vecerik et al., 2017; Nair et al., 2018; Hansen et al., 2022; Ball et al., 2023; Paine et al., 2019). While these methods accelerate learning through careful reuse or constraint of demonstration data, they often impose a tight coupling between the learning algorithm and the expert policy distribution. Instead of using expert demonstrations to constrain learning or warm-start policies, DGN obtains much more efficient learning by adaptively guiding the policy toward actions represented by the demonstrations, while still allowing the agent to discover and optimize its own reward-maximizing policy.

**Combining Imitation and Reinforcement Learning.**    One particularly common approach that leverages expert demonstrations in RL combines the objective with imitation learning (IL). A straightforward technique is to first pre-train a policy on demonstrations and then fine-tune it with RL (Silver et al., 2016; Hester et al., 2018; Rajeswaran et al., 2017). However, a challenge with this pipeline is that the policy, if optimized only with RL, may forget the initial demonstrated behaviors. To address this, many methods use an imitation loss, i.e. behavior cloning regularization, during RL to keep the policy close to the expert. DQfD (Hester et al., 2018) does this integration of IL and RL in the loss, and regularized optimal transport (ROT) adapts the weight of the imitation loss over time (Haldar et al., 2023). More recent methods have explicitly maintained both IL and RL policies. These include Policy Expansion (PEX) (Zhang et al., 2023), which uses a reference offline RL policy during online exploration, and imitation-bootstrapped RL (IBRL) (Hu et al., 2023), which first trains a separate IL policy and uses it to propose actions alongside the RL policy. In contrast to the above approaches, which often require careful tuning of loss weights or maintenance of separate policies, DGN uses demonstrations only to guide sampling in the RL process, aiming to leverage expert data without requiring an explicit imitation loss. DGN can also be integrated into existing IL + RL pipelines, as we show in our experiments.

## 3    PRELIMINARIES

We consider an online reinforcement learning (RL) setting where an agent interacts with a Markov Decision Process (MDP) defined by the tuple $(\mathcal{S}, \mathcal{A}, p, r, \gamma)$, where $\mathcal{S}$ is the set of states, $\mathcal{A}$ the set of actions, $p(s' \mid s, a)$ the transition dynamics, $r(s, a)$ the reward function, and $\gamma \in [0, 1)$ the discount factor. At each timestep $t$, the agent observes state $s_t \in \mathcal{S}$, selects an action $a_t \in \mathcal{A}$ according to a policy $\pi(a \mid s)$, receives reward $r(s_t, a_t)$, and transitions to a new state $s_{t+1} \sim p(\cdot \mid s_t, a_t)$.

The goal of the agent is to learn a policy $\pi_\theta(a|s)$ that maximizes the expected discounted return: $\max_\theta \mathbb{E}_{\pi_\theta}[\sum_{t=0}^{T} \gamma^t r(s_t, a_t)]$ where $\pi_\theta(a|s) := \mathcal{N}(\mu_\theta(s), \Sigma)$. In addition to interacting with the environment, the agent has access to an expert dataset $\mathcal{D}_{\text{data}} = \{\tau_1, \ldots, \tau_N\}$. Each trajectory $\tau_i = \{s_0, a_0, r_0, s_1, a_1, \ldots, s_T, a_T, r_T\}$ consists of a sequence of states and actions as well as sparse reward signal. This dataset is used to guide learning, particularly in the early stages when reward signals are sparse or difficult to obtain.

## 4    RL WITH DATA-GUIDED NOISE

To address the challenge of sparse rewards and sample inefficiency in online RL, our key insight is that prior data, particularly expert data, is especially valuable for identifying what kinds of exploratory actions are likely to be effective—not necessarily what the final policy should do. We introduce DGN, a framework that leverages data to guide RL with implicit imitation signals, without relying on imitation losses or constraining the policy. Our proposed framework aims to shape the agent's exploratory behavior–rather than its policy updates–using learned noise. This learned noise increases the probability of the policy taking actions that are represented in the demonstration data.

To realize the framework proposed by DGN, one practical instantiation of implicit imitation signals is learning a state-dependent zero-mean Gaussian noise distribution over the action difference between the prior demonstration data and the RL policy in the dataset states. We can view this instantiation as *learning the mean of the policy via RL and the variance via imitation for that mean.* More formally, this procedure learns a parameterized policy $\pi_{\text{sampling}}(a \mid s)$, where the action distribution is modeled as a Gaussian distribution $\mathcal{N}(\mu_\theta(s), \Sigma_\phi(s))$, to sample actions during training. Here, $\mu_\theta(s)$ is the mean of the policy learned via RL. We learn $\Sigma_\phi$ via imitation to match the action differences around

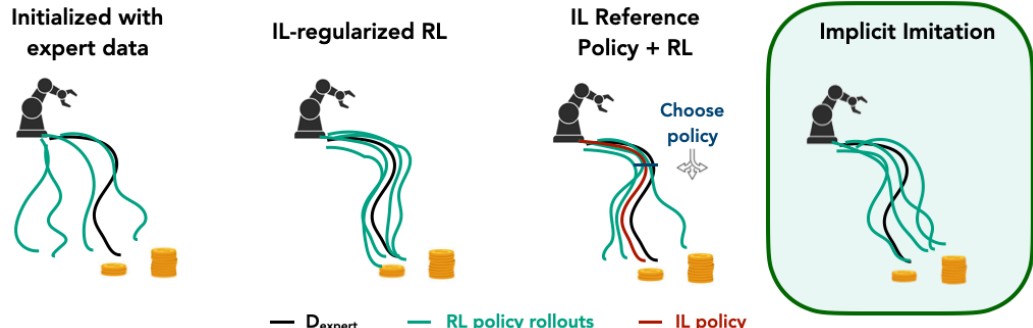

Figure 2: **Behavior of Online RL with Expert Data.** Prior work has proposed several strategies for sparse-reward RL that leverage expert data. Initializing the replay buffer with expert data does not directly use the expert information to maximally accelerate learning. In IL-regularized RL, the agent is constrained to mimic expert actions, which may limit the agent from finding more optimal solutions. IL + RL frameworks that use a reference policy rely on training a strong IL policy and robust modulation between policies. Instead of using explicit imitation constraints, DGN implicitly guides exploration by using expert-policy action differences to learn a noise distribution that accelerates the agent's learning.

the mean. In this sense, $\Sigma_\phi$ controls the structure and scale of exploration noise, so the policy learns how to act through RL and how to explore through prior data supervision.

## 4.1 Learning Data-Guided Noise

We instantiate DGN with modeling the noise as a learned, state-dependent Gaussian. The noise captures the direction and scale of the differences to the dataset distribution, increasing the probability of policy taking actions represented in the dataset without explicitly constraining the policy.

**DGN with a zero-mean Gaussian.** Let $\mu_\theta(s)$ be the mean of the current policy parameterized by $\theta$. We learn the sampling policy $\pi_{\text{sampling}}(a|s)$ to model $\mathcal{N}(\mu_\theta(s), \Sigma_\phi(s))$ corresponding to learning the mean via RL and variance via imitation. The learned state-conditioned covariance matrix $\Sigma_\phi(s)$ is parameterized by an MLP that maps from states $s$ to the Cholesky decomposition $A_\phi(s)$ of the covariance matrix $\Sigma_\phi(s) = A_\phi(s)A_\phi(s)^T$.

Our training objective for $\phi$ for the covariance matrix is to minimize the negative log-likelihood of $\pi_{\text{sampling}}(a|s)$ on $(s, a)$ pairs from $\mathcal{D}_{\text{data}}$:

$$\min_\phi \sum_{(s,a)\in\mathcal{D}_{\text{data}}} -\log\pi_{\text{sampling}}(a|s) \text{ where } \pi_{\text{sampling}}(a|s) := \mathcal{N}(\mu_\theta(s), \Sigma_\phi(s)) \tag{1}$$

Every $N$ environment steps, we fine-tune $\Sigma_\phi(s)$ using the latest RL policy $\pi_\theta(a|s)$ for each $(s, a)$ in the prior dataset $\mathcal{D}_{\text{data}}$.

**Alternative DGN Formulation.** There are other ways we can model the difference between policy actions and dataset actions as noise. One particular example is that we can fit a full residual policy via imitation learning rather than only fitting the covariance, which gives $\mathcal{N}(\mu_\phi(s), \Sigma_\phi(s))$ as the noise distribution. We let $\mu_\phi(s)$ denote a learned state-conditioned mean and $\Sigma_\phi(s)$ the covariance matrix parameterized the same way as the zero-mean formulation. This gives $\pi_{\text{sampling}}(a|s) := \mathcal{N}(\mu_\theta(s) + \mu_\phi(s), \Sigma_\phi(s))$

## 4.2 Data-Guided Perturbations for Exploration

We use the trained sampling policy to guide exploration during rollouts. Actions taken in the environment are sampled from the sampling policy which learns expert-guided perturbations. More formally, at each timestep, the agent samples an action as:

$$a_t \sim \pi_{\text{sampling}}(a_t|s_t)$$

---

**Algorithm 1** RL via Implicit Imitation (DGN)

---

**Require:** Prior dataset $\mathcal{D}_{\text{data}} = \{(s_i, a_i)\}$, initialize policy $\pi_\theta$, sampling policy $\pi_{\text{sampling}}$
1: **while** training **do**
2:     **Collect rollouts:**
3:     **for** each environment step $t$ **do**
4:         Sample $a_t$ from $\pi_{\text{sampling}}(a_t|s_t)$
5:         Take action $a_t$ and observe $r_t$ and $s_{t+1}$ from the environment
6:         Store $(s_t, a_t, r_t, s_{t+1})$ in RL replay buffer
7:     **end for**
8:     **Update policy:**
9:     Perform standard off-policy RL updates with collected data
10:     **if** time to update sampling policy **then**
11:         Update $\phi$ by minimizing negative log-likelihood:

$$\min_\phi \sum_{(s,a)\in\mathcal{D}_{\text{data}}} -\log \pi_{\text{sampling}}(a|s)$$

12:     **end if**
13: **end while**

---

Importantly, this does not constrain the policy to stay near the expert distribution: the noise distribution is only used to perturb actions during exploration, not to alter the policy optimization objective.

If the policy eventually surpasses the expert demonstrations in performance, the learned noise may remain large in magnitude, potentially pulling the agent away from its improved behavior. To mitigate this, one strategy is to apply an annealing schedule to the noise during training. This annealing schedule ensures that early in training, exploration relies on guidance by expert-informed noise, but as learning progresses and the policy improves, the influence of the noise decreases, so it eventually relies more on its own learned behavior. Specifically, we scale the sampled noise by an inverse exponential factor:

$$\tilde{\epsilon}_t = \epsilon_t \cdot \exp\left(-\frac{t}{\tau}\right),$$

where $t$ is the number of environment steps that have been taken and $\tau$ is the annealing timescale. Another strategy to allow the agent to rely on its own learned behavior after initial exploration is to turn off data-guided noise and set $\tilde{\epsilon}_t = 0$ when the last $n$ training episodes reaches $m\%$ success rate. The full algorithm is shown in Algorithm 1.

## 5   EXPERIMENTAL RESULTS

Our experimental evaluations aim to answer the following core questions:

1. Is DGN able to leverage imitation signals for improvement over standard RL and methods that use explicit imitation regularization?

2. How does DGN perform compared to methods that rely on a reference imitation learning policy?

3. What components of DGN are most important for performance?

We evaluate DGN on a set of 7 challenging continuous control tasks from Adroit and Robomimic. We present the tasks in Figure 3. The Adroit benchmark suite involves controlling a 28-Dof robot hand to perform complex tasks of spinning a pen (`pen-binary-v0`), opening a door (`door-binary-v0`), and relocating a ball (`relocate-binary-v0`). The policy needs to learn highly dexterous behavior to successfully complete each task. In the Robomimic (Mandlekar et al., 2021) environment suite, we evaluate on `Lift`, `Can`, `Square`, and `Tool Hang`, which involve controlling a 7-Dof robot arm to lift up a cube, pick up a can and place it in the correct bin, insert a tool on a square peg, and hang a tool on a rack, respectively. All environments have sparse binary rewards to signal task completion at the end of an episode. For all tasks, the observations consist of robot proprioception as well as the pose of objects in the environment relevant to the task. For Adroit,

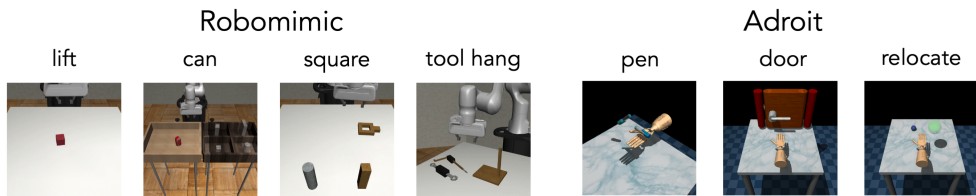

Figure 3: Visualizations of the seven environments on which we evaluate DGN.

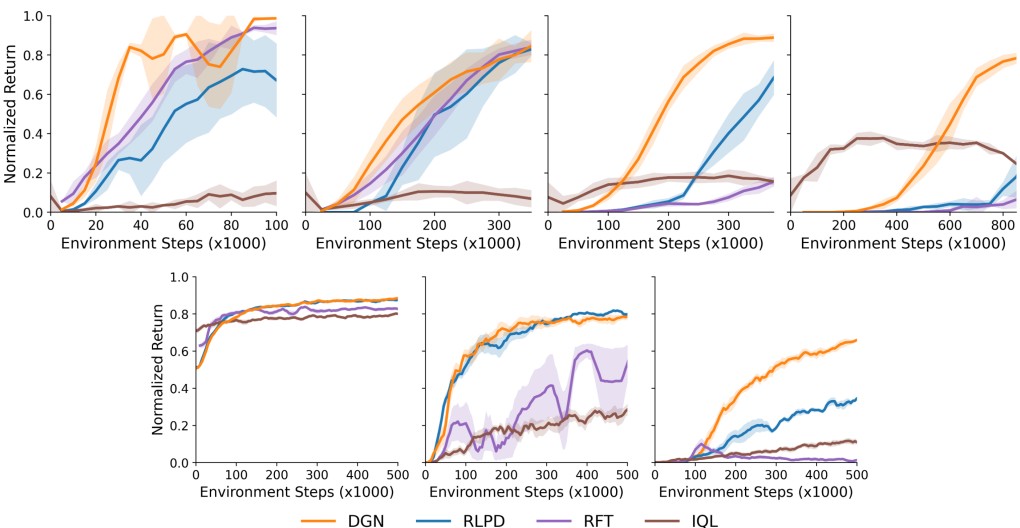

Figure 4: **Average Normalized Returns** for Robomimic and Adroit tasks comparing with standard unconstrained RL and imitation-regularized RL methods. Across all tasks, DGN consistently exceeds or matches the performance of the best baseline—even as the best baseline method varies by task. The relative benefit of DGN over RLPD and other baselines is larger on the most difficult tasks: `square`, `tool hang`, and `relocate`.

we use data from a combination of human teleoperation and trajectories collected from a BC policy. For the Robomimic tasks, we use demonstrations from the proficient-human dataset provided by the paper. Several of these tasks (e.g., `Tool Hang`, `Relocate`) are known to be especially difficult for standard RL algorithms under sparse rewards.

We evaluate DGN against four state-of-the-art comparisons that leverage prior data for training: (1) RL with Prior Data (**RLPD**) (Ball et al., 2023), which initializes the replay buffer with prior data and oversamples from it for online training; (2) Regularized Fine-Tuning (**RFT**), which pretrains the policy with imitation learning and adds a imitation learning loss to the RL objective with a regularization weight, encouraging the policy to remain close to the prior data throughout training. (3) Implicit Q-Learning (**IQL**) (Kostrikov et al., 2021) finetuning, which learns a value function using expectile regression and a policy via advantage-weighted regression that constrains the policy to be close to behavior data. (4) Imitation-Bootstrapped RL (**IBRL**) (Hu et al., 2023), which first trains an IL policy with the offline data and then chooses between actions proposed by the IL and RL policies using a Q-function.

We instantiate DGN on top of RLPD, with the only changes being the guided sampling model described in Section 4. The state-dependent learned covariance MLP is trained every $N = 1000$ environment steps on Robomimic tasks and every $N = 2000$ steps on Adroit tasks. For the Adroit experiments, we anneal perturbations with $\tau = 30000$ and we turn off noise for the Robomimic environments when the previous $n = 10$ training episodes reaches $m = 50\%$ success rate. We report the average and standard error across three runs for all experiments.

## 5.1 Does DGN improve over standard RL and imitation-regularized RL methods?

For the first set of experiments, we compare DGN against unconstrained RL algorithms and algorithms that utilize explicit imitation regularization across Adroit and Robomimic benchmarks. We focus on two questions, is DGN able to leverage imitation signals to outperform unconstrained RL with prior data approaches, and how does the implicit imitation signal from DGN compare with explicit regularization approaches. We present the results in Figure 4. First, we find that DGN outperforms or matches the performance of RLPD on every task. In particular, on the harder tasks `relocate` and `tool hang`, DGN outperforms RLPD by a significant margin and requires significantly fewer samples to learn to solve the task. While RLPD utilizes prior data, the policy is not directly influenced by it, so it is not able to maximally get the benefit of sample efficiency from imitation signals. In contrast, the implicit imitation signals greatly accelerate policy learning by guiding the policy to explore in expert-like directions. Even on the Adroit tasks which uses a mix of human teleoperation and IL policy data, DGN was able to outperform RLPD by 2x on the hardest `relocate` task. Comparing with RFT and IQL, which are two approaches for explicitly constraining the policy to imitation signals, we see that DGN outperforms them even more as the imitation learning regularization does not align with reward maximization of RL. This is particularly clear in `door` as the policy alternates between improving and zero performance. While DGN provides implicit imitation signals, it does not force RL to balance between losses, which allows it to find its own high reward policy while getting benefits from the imitation signal.

## 5.2 How does DGN perform compared to methods that rely on a reference imitation policy?

We next compare DGN to IBRL, a state-of-the-art reference policy based approach. While IBRL trains an unconstrained RL policy for reward maximization, it relies on a strong IL policy to derive benefits. We make the comparison of DGN with IBRL under three conditions, with a high-quality BC policy trained to convergence, a low-quality BC policy obtained by underfitting the data, and using a multimodal dataset. To stay consistent with IBRL implementation details, we apply dropout to the actor. We find that this generally helps the performance of DGN, but it is a design choice orthogonal to taking advantage of implicit imitation signals so we only include it for experiments in this section.

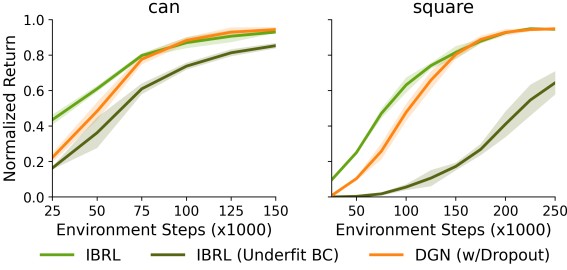

Figure 5: **Comparison to IBRL.** Average normalized returns on `can` and `square`, comparing to the reference policy-based approach of IBRL. IBRL's performance strongly depends on having a well-trained IL policy, and its performance can degrade substantially without it, while DGN does not.

**Varying the quality of the IL policy:** As seen in Figure 5, across `can` and `square`, DGN performs comparable to IBRL even when IBRL has access to a strong BC policy. This indicates implicit imitation signals alone are enough to derive full benefits of imitation signals, without the need to train a reference policy. However, we find that IBRL's performance degrades significantly without a strong IL policy e.g., due to insufficient training time or limited capacity, and this is especially the case for the harder task of `square`, causing it to have substantially lower success. This is because IBRL's ability to leverage imitation signals is dependent on the IL policy capturing the right distribution, which can be a significant assumption to make in practice. In contrast, DGN does not suffer from this problem as we leverage imitation signals only for guiding sampling.

**Using a multimodal dataset:** IBRL uses a reference IL policy to guide RL exploration. The effectiveness of this approach depends heavily on the quality of the IL policy. We further evaluate IBRL on a more challenging setting: a multimodal dataset composed of successful trajectories. We defer to Appendix A for details on the setup. Despite all trajectories being successful demonstrations, the diversity of strategies introduces multimodality that makes learning a reliable IL policy difficult.

In Figure 6, we see that the performance of IBRL degrades significantly when the data is multimodal, and this is apparent on both `can` and `square`, even when we carefully train the reference IL policy. In contrast, DGN appears less sensitive to mode quality. Because it does not rely on executing a fixed imitation policy, DGN can still extract useful exploratory structure from the expert data.

### 5.3    WHAT COMPONENTS OF DGN ARE MOST IMPORTANT FOR PERFORMANCE?

To better understand the importance of different components of DGN for policy performance, we ablate over two key components that could affect the performance of DGN: learning a full residual policy via imitation learning rather than only the covariance and the state-dependence of the learned covariance.

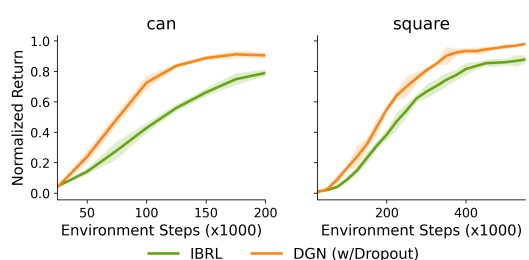

**Ablation on learning a full residual policy via imitation learning:**    A key design choice in DGN is how the exploration noise is modeled. We consider two variants of DGN in this ablation: one where only the covariance of $\pi_{\text{sampling}}$ is learned via imitation, and another where the mean is learned alongside the covariance.

Figure 6: **Comparison to IBRL with a multimodal dataset.** DGN outperforms IBRL, showing DGN is less sensitive to dataset quality and multimodality.

Empirically, we find that both variants lead to comparable performance across the Robomimic tasks, as seen in Figure 7. This suggests that the benefits of DGN are not tied to whether the mean is fixed at zero or learned — rather, they stem from the broader mechanism of using expert-policy differences to inform the structure of the noise to shape the exploration distribution. This highlights that the core strength of DGN lies not in enforcing imitation, but in extracting exploration priors from demonstrations that help RL discover useful behaviors more efficiently.

**Ablation on state-conditioning:**    An important component of DGN is that the learned data-guided noise is conditioned on the current state. This state-conditioning allows the exploration noise to be adapted dynamically, capturing task-specific differences in how expert actions deviate from policy actions across the state space.

To isolate the importance of state-conditioning the distribution, we test an ablation of DGN where we learn a covariance matrix in the same way as DGN, but this covariance matrix is no longer state-conditioned. We replace the learned sampling policy $\pi_{\text{sampling}}(a|s) := \mathcal{N}(\mu_\theta(s), \Sigma_\phi(s))$ with $\pi_{\text{sampling}}(a|s) := \mathcal{N}(\mu_\theta(s), \Sigma_\phi)$. The covariance matrix $\Sigma_\phi$ is now parameterized by single matrix of learned weights $A_\phi$, representing the Cholesky decomposition of $\Sigma_\phi$. This matrix of parameters is learned the same way as DGN's state-dependent covariance matrix, following equation 4.1, with the important difference that the parameters are no longer state-dependent.

The results in Figure 8 show that this ablation of DGN without state-conditioning of the covariance matrix substantially underperforms DGN. The performance drop is especially apparent in `can`, and

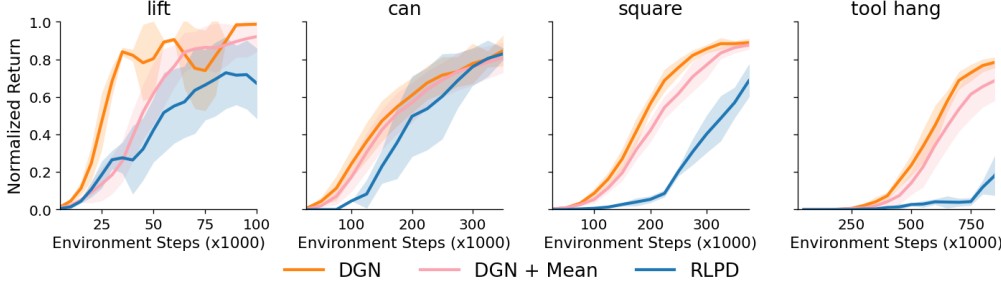

Figure 7: **Ablation on Learning Full Residual Policy via Imitation Learning**. Learning a full residual policy via imitation performs similarly to only learning the covariance.

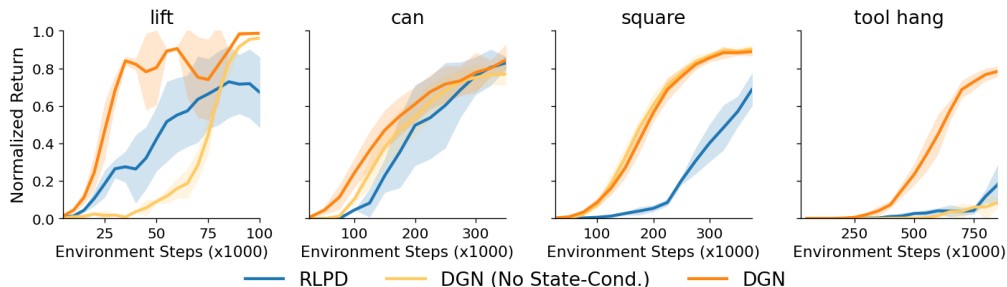

Figure 8: **Ablation on State-Conditioning DGN Distribution**. The ablation of DGN without state-conditioning of the learned covariance matrix performs worse than DGN on Robomimic tasks.

`tool hang`, where vanilla RLPD outperforms the ablation with no state conditioning, indicating the importance of learning state-dependent noise.

**How does the action distribution of DGN compare to other methods?** To further understand why the implicit imitation signals of DGN could be significantly more preferable than explicit regularization, we analyze the KL divergence between the action distributions of each policy and that of an IL policy trained on demonstrations over a set of demonstration states. As shown in Figure 9, while all methods initially reduce their KL divergence, as they all learn to imitate demonstrator behavior early, the plot shows differences in how each method balances imitation and reward maximization. Our method maintains a higher KL divergence from the IL policy than RFT and IBRL throughout training. This reflects greater freedom to deviate from the demonstrations for better reward optimization.

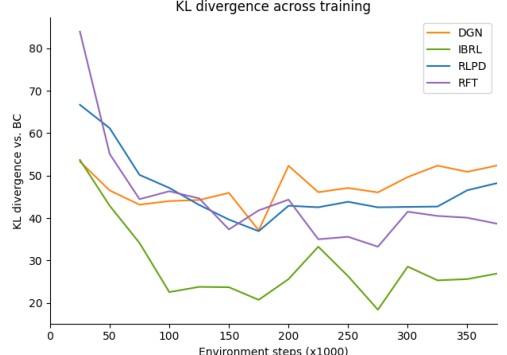

Figure 9: **KL Divergence from BC Policy over Training.** On the `square` task, we plot the KL divergence between each method's policy and a BC policy trained on expert demonstrations, evaluated on a fixed set of demonstration states. All methods initially reduce their divergence, reflecting early-stage imitation. However, DGN maintains a consistently higher divergence than IBRL and RFT throughout training.

## 6 LIMITATIONS AND CONCLUSION

In this work, we introduced DGN, a framework that leverages prior data such as expert demonstrations not by constraining the policy through explicit imitation, but by shaping the agent's exploration behavior through implicit imitation in the form of prior data-guided noise. Using the state-dependent differences between expert and policy actions, DGN injects structured noise into action selection, encouraging the agent to explore in directions that align with successful behaviors. This approach enables efficient reward discovery early in training while allowing the policy to improve beyond the demonstrations. Our experiments across a range of challenging sparse-reward continuous control tasks demonstrate that DGN consistently matches or outperforms both standard RL and imitation-augmented methods.

Despite these promising results, limitations remain that suggest avenues for future work. In particular, while our framework is general for any approach that learns an implicit imitation signal from prior data to guide the policy, we explore one specific instantiation as a state-dependent Gaussian distribution. It would be interesting to study how different modeling choices and sampling strategies impact performance to understand what would be the best instantiation of our proposed framework.

## REPRODUCIBILITY STATEMENT

We fully describe the method and our algorithm in the main paper text. We include a table of hyperparameters we used and describe other experimental details in the appendix. Additionally, we attach the code we ran to produce the presented results. The code submission includes configuration files, and instructions to download the data and run the code are included in the README files in the code.

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
