# OpenReview forum: "Reinforcement Learning via Implicit Imitation Guidance"
_ICLR.cc/2026/Conference — Submitted to ICLR 2026_

### Official Review · Reviewer_zeD3 · 2025-10-25

**Soundness:** 2
**Presentation:** 3
**Contribution:** 2
**Rating:** 4
**Confidence:** 3

**Summary:**

This work tackles sample-efficient reinforcement learning using prior data without relying on imitation learning objectives that may hinder long-term reward optimization. The proposed Data-Guided Noise (DGN) framework leverages demonstrations to guide exploration through data-informed policy noise, rather than enforcing behavior cloning. This approach yields 2–3× performance gains over previous RL-from-offline-data methods across seven continuous control tasks.

**Strengths:**

- The writing is clear and easy to follow, and the illustrations effectively convey the proposed method.

- The proposed approach serves as a flexible framework that integrates naturally with other imitation learning for RL methods and can be easily combined with various RL algorithms.

**Weaknesses:**

- The paper lacks discussion and comparison with online imitation learning methods (e.g., [1,2,3]). These methods operate under a harder setting using expert datasets with reward free interactions, whereas this work assumes access to both expert data and reward signals. The authors should include a discussion and empirical comparison with this line of research to better highlight the advantages of their imitation guided RL framework.

- In Figure 9, the authors argue that their method achieves a larger KL divergence compared to prior approaches, implying greater flexibility in policy optimization. However, a larger KL divergence does not necessarily indicate better policy performance, especially if the optimal policy is expected to be close to the behavior cloning (BC) policy. The primary benefit of incorporating online RL over pure BC lies in improving policy coverage, yet no experiments are provided to evaluate performance on out-of-distribution (O.O.D.) states beyond the expert demonstrations, leaving this claimed advantage unsubstantiated.


[1] Ren, J., Swamy, G., Wu, Z. S., Bagnell, J. A., & Choudhury, S. (2024). Hybrid inverse reinforcement learning. arXiv preprint arXiv:2402.08848.

[2] Yin, Z. H., Ye, W., Chen, Q., & Gao, Y. (2022). Planning for sample efficient imitation learning. Advances in Neural Information Processing Systems, 35, 2577-2589.

[3] Li, S., Huang, Z., & Su, H. (2025). Reward-free World Models for Online Imitation Learning. In Forty-second International Conference on Machine Learning.

**Questions:**

- Could the authors provide a more detailed discussion and comparison with existing online imitation learning literature to better position their work within this research landscape?

- Could the authors add experiments on out-of-distribution (O.O.D.) samples to demonstrate the advantages of incorporating online RL over purely offline imitation learning?

---

> ### Author Response · Authors · 2025-12-02
>
> Dear reviewer zeD3,
>
> Thank you for your thoughtful feedback. We address the points raised individually below.
>
> > In Figure 9, the authors argue that their method achieves a larger KL divergence compared to prior approaches, implying greater flexibility in policy optimization. However, a larger KL divergence does not necessarily indicate better policy performance, especially if the optimal policy is expected to be close to the behavior cloning (BC) policy. The primary benefit of incorporating online RL over pure BC lies in improving policy coverage, yet no experiments are provided to evaluate performance on out-of-distribution (O.O.D.) states beyond the expert demonstrations, leaving this claimed advantage unsubstantiated.
>
> We argue that a larger KL divergence indicates that DGN does not suffer from forcing the RL policy to be too close to the IL policy even when it is not optimal to do so. While the expert demonstrations are a helpful guide, we work in a setting where IL alone would not achieve satisfactory performance because a pure-IL policy would either 1) overfit to the expert dataset, which has insufficient state coverage and number of samples to train an IL policy through pure IL, or 2) learn good, but still suboptimal, behaviors demonstrated by the expert. Additionally, for some of the baseline methods, there can be a clash between the IL regularization and RL actor objectives, which can degrade policy performance relative to using one of these objectives alone. The higher KL divergence suggests DGN is less tightly bound to the IL policy, which suggests DGN can still benefit from faster convergence with the aid of expert demo guidance without suffering from these issues.
>
> We agree that the included KL divergence plot does not speak to behavior on OOD states directly, but may do so indirectly, as it may reflect less overfitting to the expert dataset and the state space it covers due to the presence of broader state space coverage in the dataset. Our policies are evaluated by rolling out the learned RL policy. In this process, the policy will encounter states outside the state distribution of the expert dataset.
>
> > The paper lacks discussion and comparison with online imitation learning methods (e.g., [1,2,3]). These methods operate under a harder setting using expert datasets with reward free interactions, whereas this work assumes access to both expert data and reward signals.
>
> We believe that such methods are designed for a different problem setting. While it would be exciting to see future work apply DGN to problem settings where the RL agent does not have access to reward signals, we believe that this is outside the scope of this work.

---

### Official Review · Reviewer_FcCQ · 2025-10-28

**Soundness:** 2
**Presentation:** 3
**Contribution:** 2
**Rating:** 2
**Confidence:** 2

**Summary:**

The paper focuses on applying reinforcement learning (RL) in the continuous control setting, where the agent is further provided some expert demonstrations.
The paper claims that simply including the demonstrations into a replay buffer, or using a imitation-learning (IL) regularizer utilizes the demonstrations ineffectively.
Instead, the paper proposes to parameterize the exploration Gaussian policy such that its covariance is computed based on the demonstrations.
The paper evaluates the proposed method, DGN, on continuous-control tasks from Adroit and Robomimic in the sparse reward setting, and show that DGN outperforms, if not comparable, against three existing RL algorithms that utilizes demonstrations.

**Strengths:**

- The paper is easy to read and the idea is simple
- The performance outperforms compared approaches, and the paper provides some ablation studies on a subset of design choices

**Weaknesses:**

- Related work, combining imitation and reinforcement learning: I believe there is a line of research that uses hierarchical (inverse) RL [1-4] to better explore the environment. It would be nice if the paper includes some discussions on these papers as well.
- Method
	- In section 4.1, it seems like the method will encourage the agent to explore indefinitely as the variance will be large asymptotically when the "expert" policy is suboptimal and the RL policy is optimal, or when there are multiple expert actions from the same states. Based on the training/evaluation split this might be okay as during evaluation we use the greedy policy, however practitioners applying RL will A/B test policies along with the training phase.
	- In section 4.2, the paper argues that IL regularizer will restrict the learner from learning the optimal policy which I agree. However, the paper alleviates the above problem using a manually defined hyperparameter decaying schedule, which arguably, can also be applied to the IL regularizer.
		- Furthermore, the schedule choice seems arbitrary, in the sense that they could be using linear schedule, cosine schedule, etc.
- Experiments
	- Nit: Figure 4 should include the environment name per plot.
	- Regarding the comparison against IBRL on line 370---I have trouble agreeing with the argument that IL policy has difficulty capturing the right distribution. Nowadays, BC policies are trained with diffusion policies (DPs), which can capture multimodal distributions [5]. Furthermore, it has been shown that the compounding error can be alleviated with DPs [6]. Finally, there are works recently [7, 8] that focuses on policy steering in DPs and show promising results, which I think this work should also compare against.
	- The choice of hyperparameters might seem arbitrary for the scheduling choice, as well as $\tau, n, m$.
	- I think there can be analysis on checking how to covariance changes over time, and how that dynamics change based on the modality of the dataset.

References:
[1] Riedmiller, Martin, et al. "Learning by playing solving sparse reward tasks from scratch." International conference on machine learning. PMLR, 2018.
[2] Hertweck, Tim, et al. "Simple sensor intentions for exploration." arXiv preprint arXiv:2005.07541 (2020).
[3] Ablett, Trevor, Bryan Chan, and Jonathan Kelly. "Learning from guided play: Improving exploration for adversarial imitation learning with simple auxiliary tasks." IEEE Robotics and Automation Letters 8.3 (2023): 1263-1270.
[4] Ablett, Trevor, et al. "Efficient Imitation Without Demonstrations via Value-Penalized Auxiliary Control from Examples." 2025 IEEE International Conference on Robotics and Automation (ICRA). IEEE, 2025.
[5] Chi, Cheng, et al. "Diffusion policy: Visuomotor policy learning via action diffusion." The International Journal of Robotics Research 44.10-11 (2025): 1684-1704.
[6] Zhang, Thomas T., et al. "Imitation Learning in Continuous Action Spaces: Mitigating Compounding Error without Interaction." arXiv preprint arXiv:2507.09061 (2025).
[7] Wagenmaker, Andrew, et al. "Steering Your Diffusion Policy with Latent Space Reinforcement Learning." arXiv preprint arXiv:2506.15799 (2025).
[8] Wang, Yanwei, et al. "Inference-time policy steering through human interactions." 2025 IEEE International Conference on Robotics and Automation (ICRA). IEEE, 2025.

**Questions:**

See above.

---

> ### Author Response · Authors · 2025-12-02
>
> Dear reviewer FcCQ,
>
> Thank you for your thoughtful points of feedback. We address them as follows.
>
> > there is a line of research that uses hierarchical (inverse) RL [1-4] to better explore the environment. It would be nice if the paper includes some discussions on these papers as well.
>
> Thanks for pointing out the connection to this line of work. We will include a discussion of this line of work in the related works section.
>
> > In section 4.1, it seems like the method will encourage the agent to explore indefinitely as the variance will be large asymptotically when the "expert" policy is suboptimal and the RL policy is optimal, or when there are multiple expert actions from the same states. Based on the training/evaluation split this might be okay as during evaluation we use the greedy policy, however practitioners applying RL will A/B test policies along with the training phase.
>
> While the variance after running DGN for a long time will be larger than its minimum when the RL policy aligned with the expert, it should not be very large at this time, as the expert should not differ tremendously from an optimal policy. Additionally, the shutoff or annealing mechanism of DGN will make the variance vanish asymptotically. By the time a policy is good enough to start A/B testing alongside training as described, the learned variance may be shut off.
>
> > the paper argues that IL regularizer will restrict the learner from learning the optimal policy…However, the paper alleviates the above problem using a manually defined hyperparameter decaying schedule, which…can also be applied to the IL regularizer
>
> We ran RFT for the square and tool-hang tasks with several annealing schedules and found that DGN substantially outperforms each of these. This suggests that simply annealing the BC regularization of RFT does not work as well as DGN. Additionally, the hyperparameters of DGN’s shutoff mechanism do not require task-specific tuning. With the shutoff mechanism,the learned covariance matrix is no longer used for exploration after the success rate hits 50 percent, allowing the mechanism to adjust automatically to varying task difficulty.
>
> | Mean Success Rate | Square (Step 350K) | Tool Hang (Step 850K) |
> |-------------------|-------------------:|----------------------:|
> | DGN               | 88 ± 2             | 78 ± 3                |
> | RFT (No Decay)    | 10 ± 3             | 8 ± 5                 |
> | RFT (T=100K)      | 3 ± 2              | 0 ± 0                 |
> | RFT (T=200K)      | 7 ± 3              | 3 ± 7                 |
> | RFT (T=300K)      | 4 ± 1              | 15 ± 17               |
> | RFT (T=400K)      | 5 ± 1              | 0 ± 0                 |
>
> > Nit: Figure 4 should include the environment name per plot.
>
> Thank you for pointing this out. We have added them.
>
> > Regarding the comparison against IBRL...I have trouble agreeing with the argument that IL policy has difficulty capturing the right distribution. Nowadays, BC policies are trained with diffusion policies (DPs), which can capture multimodal distributions [5]. Furthermore, it has been shown that the compounding error can be alleviated with DPs [6].
>
> The performance of an IL policy will depend on the size of the policy. Using a large IL policy, particularly a diffusion policy that requires multiple inference steps, may make inference too slow for some tasks. Additionally, even if an IL policy can capture the expert distribution, in general, this distribution will not precisely match the distribution that maximizes RL’s reward-maximizing objective, which is the relevant distribution when using an IL policy to help RL.
>
> > I think there can be analysis on checking how to covariance changes over time, and how that dynamics change based on the modality of the dataset.
>
> We have attached a table showing the average magnitude of the noise sampled using the learned covariance matrix (smoothed with a median taken over a sliding window) for the square task. The magnitude of the noise sampled from the covariance matrix decreases over time. Intuitively, this makes sense since the gap between the expert and RL policy actions decreases as the RL policy improves.
>
> | Step (x1000) | Smoothed Action Permutation Magnitude |
> |--------------|---------------------------------------|
> |            0 | 0.566 ± 0.002                         |
> |           25 | 0.406 ± 0.042                         |
> |           50 | 0.298 ± 0.018                         |
> |           75 | 0.273 ± 0.013                         |
> |          100 | 0.292 ± 0.0194                        |
> |          125 | 0.173 ± 0.078                         |

---

### Official Review · Reviewer_u3Py · 2025-11-01

**Soundness:** 4
**Presentation:** 2
**Contribution:** 3
**Rating:** 4
**Confidence:** 4

**Summary:**

The paper proposes Data-Guided Noise (DGN), a method to improve sample efficiency in reinforcement learning by using prior demonstration data. Instead of using data for explicit imitation learning objectives, DGN learns a state-conditioned noise distribution based on the difference between expert actions and the current policy's actions. This noise is used to guide exploration during training.

**Strengths:**

- The paper proposes a novel method for leveraging demonstration data to guide RL exploration by shaping the noise distribution, rather than through an explicit imitation loss.

- The empirical evaluation is thorough. The method is compared against several relevant baselines (RLPD, RFT, IQL, IBRL), and the core components of the DGN framework are carefully validated through various ablation studies.

**Weaknesses:**

- **Contextualization w.r.t. related work**: The paper's discussion of combining Imitation Learning (IL) and Reinforcement Learning (RL) primarily focuses on two strategies: (1) combining IL/RL objectives and (2) using a separate IL policy to guide or propose actions. However, there are a couple other approaches for guiding the exploration of the RL agent with demonstrations while avoiding the need to balance between explicit IL/RL objectives, or incurring state-action distribution shift issues experienced by replay buffer approaches. Further, there is a long line of work in robotics on residual-based control where, where RL is used to train a residual policy to improve pre-existing controllers. None of these lines of work were mentioned, and I think discussing them would improve contextualization w.r.t. related work.

  - **Guiding RL exploration with demonstrations**:

    - [Salimans & Chen 2018](https://arxiv.org/abs/1812.03381) consider resetting to expert states from a single demonstration in reverse order, to guide RL exploration in sparse reward, goal-reaching tasks.

    - [Wang et al. 2023](https://arxiv.org/abs/2210.14428)  formulates a potential-based shaped reward that treats demonstration states as goals to inform RL exploration while retaining the ability to discover the optimal policy w.r.t. the task reward

  - Using RL to learn residual policies on top of pre-existing controllers:

    - [Kasaei et al. 2023](https://www.frontiersin.org/journals/robotics-and-ai/articles/10.3389/frobt.2023.1004490/full) uses RL to learn a residual policy to improve simulated humanoid robot locomotion for Robocup 3dSim

    - [Yang et al. 2023](https://proceedings.mlr.press/v211/yang23b/yang23b.pdf)  uses RL to learn a residual policy for learning jumping behavior for a quadruped robot

    - [Li et al. 2025](https://arxiv.org/pdf/2509.20696) learns such a residual policy to improve locomotion of a Unitree G1 humanoid

  - **Empirical improvement**: The empirical advantage of DGN over baselines is not consistently strong. In 4/7 tasks in Figure 4, DGN shows minimal or no significant improvement in sample efficiency or asymptotic return in several of the tasks. While achieving SOTA results isn’t a requirement for a paper to be accepted to ICLR, the strength of the results in this paper will naturally limit the impact/interest from the ICLR community.

  - **Argument for DGN’s advantages over related methods is weak**: The paper argues that the common strategy of regularizing the RL objective with an IL objective has the following problems: (1) it requires carefully tuning the loss weights (Line 127), and (2) it constrains the policy to remain close to the expert distribution, even when no longer needed (Line 37). The paper then argues that DGN is a novel strategy for guiding RL with IL that doesn’t suffer these issues (Line 46), but I am not convinced that this is the case.

    - In Section 4.2, the authors state, “If the policy eventually surpasses the expert demonstrations in performance, the learned noise may remain large in magnitude, potentially pulling the agent away from its improved behavior. To mitigate this, one strategy is to apply an annealing schedule to the noise during training.” (pg 5).

    - So, although DGN is not directly doing IL, DGN also requires carefully tuning the tradeoff between RL and “IL” losses via the annealing schedule. This annealing schedule seems to be just another form of tradeoff parameter that requires careful tuning, similar to the loss weights DGN claims to avoid.

**Questions:**

Questions:

- Is there any reason to prefer DGN over RLPD? The answer in the paper is that “Initializing the replay buffer with expert data does not directly use the expert information to maximally accelerate learning” but the empirical improvement of DGN is not large either.

- Could the authors compare against a residual policy learning baseline, as mentioned in the weaknesses? This approach seems directly applicable to this paper's setting and does not appear to require additional assumptions. To clarify any potential misunderstanding, this is different from the "Alternative DGN Formulation," which learns a residual mean $\mu_{\phi}(s)$ for the noise distribution. A residual policy in this paper’s setting would involve learning a policy that outputs an action residual added to the mean of a fixed IL policy.


Minor comments:

- Fig 4 seems to be missing the task name for each subplot.

---

> ### Author Response · Authors · 2025-12-02
>
> Dear reviewer u3Py,
>
> Thank you for your thoughtful feedback. Below, we provide responses to each of the points you raise.
>
> >  The empirical advantage of DGN over baselines is not consistently strong…Is there any reason to prefer DGN over RLPD?
>
> Our results show that DGN increases performance relative to RLPD more on the harder tasks that are longer-horizon and require more precision. This result holds up on both the robomimic and adroit suite of tasks and can be seen in Figure 4, where tasks from each suite are listed from left to right in ascending order of difficulty. The case for using DGN over RLPD is that it never seems to hurt performance and sometimes helps substantially, particularly on harder tasks where strong performance is harder to achieve.
>
> > there are a couple other approaches for guiding the exploration of the RL agent with demonstrations while avoiding the need to balance between explicit IL/RL objectives, or incurring state-action distribution shift issues experienced by replay buffer approaches.
>
> We appreciate your raising these other lines of work. We plan to add a discussion of them to the related works section. One key shortcoming of methods involving resets to expert states is that, while the simulated environments can be reset to states in the dataset, this is not practical in real-world environments.
>
> > So, although DGN is not directly doing IL, DGN also requires carefully tuning the tradeoff between RL and “IL” losses via the annealing schedule. This annealing schedule seems to be just another form of tradeoff parameter that requires careful tuning, similar to the loss weights DGN claims to avoid.
>
> We have run RFT for the square and tool-hang tasks with several annealing schedules (decaying the BC loss weight with an exponential schedule with timescale $T$) and find that DGN substantially outperforms each of these. This suggests that it is more difficult to tune the loss weights for RFT than for DGN. Additionally, the hyperparameters of DGN’s shutoff mechanism do not require task-specific tuning: it just entails no longer using the learned covariance matrix for exploration after the success rate hits 50 percent, allowing the mechanism to adjust automatically to varying task difficulty.
>
> | Mean Success Rate | Square (Step 350K) | Tool Hang (Step 850K) |
> |-------------------|-------------------:|----------------------:|
> | DGN               | 88 ± 2             | 78 ± 3                |
> | RFT (No Decay)    | 10 ± 3             | 8 ± 5                 |
> | RFT (T=100K)      | 3 ± 2              | 0 ± 0                 |
> | RFT (T=200K)      | 7 ± 3              | 3 ± 7                 |
> | RFT (T=300K)      | 4 ± 1              | 15 ± 17               |
> | RFT (T=400K)      | 5 ± 1              | 0 ± 0                 |

---

### Official Review · Reviewer_sd6c · 2025-11-01

**Soundness:** 3
**Presentation:** 4
**Contribution:** 2
**Rating:** 4
**Confidence:** 5

**Summary:**

This paper addresses the challenge of sample-efficient reinforcement learning in sparse-reward environments by leveraging prior demonstration data. The authors argue that traditional methods, which use explicit imitation learning losses or constraints, can overly restrict the policy and hinder long-term reward maximization. As an alternative, they propose Data-Guided Noise (DGN), a framework that uses demonstrations to guide exploration "implicitly." The core idea is to learn a state-dependent exploration noise distribution. Specifically, the mean of the policy is learned via a standard RL objective, while the covariance of the exploration noise is learned via an imitation objective that models the difference between the expert's actions and the current policy's actions.

**Strengths:**

*   The core motivation is strong and well-articulated. The idea of using demonstrations to guide exploration without being rigidly constrained by them is a compelling direction for combining imitation and reinforcement learning.
*   The paper's choice of baseline algorithms is comprehensive and appropriate. It compares against methods that represent key paradigms in the offline-to-online and imitation-augmented RL space (e.g., replay buffer initialization, explicit regularization, and reference policies).
*   The empirical results presented are impressive. DGN consistently outperforms strong baselines across multiple difficult, sparse-reward tasks, suggesting the practical effectiveness of the proposed method.

**Weaknesses:**

*   The central mechanism of the paper lacks a clear theoretical or intuitive justification. It is not immediately obvious why the difference between the expert action and the current policy's mean action should define an optimal exploration distribution. While the results are strong, the paper would be more impactful if it provided more insight into *why* this specific formulation of exploration noise is so effective.
*   The paper makes strong claims about the failure modes of imitation-based methods (e.g., they "degrade long-term performance") without providing concrete, illustrative examples to substantiate them. A simple experiment showing this degradation would make the motivation for DGN much clearer.
*   The analysis seems limited to high-quality, expert demonstrations. It is unclear how DGN would perform with suboptimal or noisy demonstration data, which is a common real-world scenario. If the noise is guided by bad data, it could potentially be detrimental to exploration.
*   The experimental evaluation, while strong, could be more thorough. The policies are trained for a relatively small number of timesteps, and it's unclear if they have fully converged. Furthermore, the plots do not specify the measures of centrality and spread (e.g., mean and standard deviation/error over seeds).

**Questions:**

*   Could the authors provide more intuition or a theoretical justification for why modeling the variance of exploration noise based on the difference between expert and policy actions is an effective strategy? For instance, why should a large deviation between the policy and the expert imply a need for high-variance exploration in that direction?
*   How is DGN expected to behave when provided with suboptimal or mixed-quality demonstrations? Does the "guidance" become detrimental in such cases, and could the annealing schedule mitigate this?
*   The paper states that explicit imitation constraints "degrade long-term performance." Could you provide a specific, perhaps toy, example where this happens and illustrate how DGN avoids this failure mode?
*   In Algorithm 1, the action `at` is sampled from `π_sampling`, which already includes the learned noise. The algorithm as written does not show the base policy's mean action being computed first. Could you clarify how the base policy `πθ` is used in the action selection process during rollouts?
*   The research questions listed in Section 5 seem somewhat incomplete. Was the potential to learn from suboptimal demonstrations considered? What about the sample complexity of learning the covariance matrix itself?
*   What are the measures of centrality and spread for the plots in Figure 4 (e.g., mean and standard deviation over seeds)? Also, was there a reason for the specific training horizon chosen, and do the performance trends hold if trained for longer?
*   Figure 2 provides a helpful conceptual overview but does not appear to be referenced in the main body of the text. Could this be added?
* The current method guides exploration based on the difference between expert and policy actions, treating all expert data equally. Have the authors considered a reward-weighted formulation? For example, could the covariance learning objective be modified to give more weight to action differences from high-reward trajectories? This seems like a more direct way to encourage exploration in promising directions and might be more robust when learning from mixed-quality data.

---

> ### Author Response · Authors · 2025-12-02
>
> Dear reviewer sd6c,
>
> Thank you for your thoughtful feedback. We answer individual points below.
>
> > Could the authors provide more intuition or a theoretical justification...why should a large deviation between the policy and the expert imply a need for high-variance exploration in that direction?
>
> Consider a robot with a parallel-jaw gripper right before it picks up a sphere whose action space is EE-poses or delta-EE-poses. If the actions the RL agent would take on the states in the expert trajectory differ from the actions the expert took in those states substantially in the x-position component of the action but only a very small amount in the y-position component, then when exploring during rollouts, it makes sense to add more exploration noise to actions in the x-position component than the y-position component shortly before grasping the sphere. In short, a state-conditioned prediction of the scale of the difference between expert and RL policy actions in each direction is a useful indicator of how broadly to explore in each direction. DGN captures this “scale of difference…in each direction” in the learned covariance matrix.
>
>
> > The paper states that explicit imitation constraints "degrade long-term performance." Could you provide a specific, perhaps toy, example where this happens and illustrate how DGN avoids this failure mode?
>
> In several environments tested in our paper, RFT, the baseline chosen to be representative of the IL-regularized methods exhibit this failure mode, often does well at the start of a training run but does poorly later on, particularly for harder tasks. The resulting policies often converge to lower performance and/or take longer to converge. This issue has been discussed in several works before; we have added a citation to this work to the sentence that makes this claim in the introduction.
>
> > How is DGN expected to behave when provided with suboptimal or mixed-quality demonstrations? Does the "guidance" become detrimental in such cases, and could the annealing schedule mitigate this?
>
> We include experiments with suboptimal “worse mh” of multimodal, low-quality demonstrations on robomimic tasks (see Figure 6) and find that DGN still outperforms IBRL. Since DGN learns a distribution that models expert-RL policy action differences, when those differences are broader on a lower-quality dataset, the sampling policy’s distribution gets broader rather than collapsing to an incorrect fixed imitation policy. While this means exploration is targeted less precisely, DGN still improves exploration and results in faster policy convergence and better performance.
> Additionally, in experiments comparing the performance of DGN and RLPD with non-expert datasets on the most difficult Antmaze environments, DGN exhibits higher sample efficiency than RLPD.
>
> | Mean Return by Step      |      |          |           |           |           |
> |--------------------------|------|----------|-----------|-----------|-----------|
> | Task                     |      | Step 50K | Step 100K | Step 150K | Step 200K |
> | antmaze-large-play-v2    | DGN  |     27.7 |      77.3 |      81.0 |      88.7 |
> |                          | RLPD |     52.0 |      73.7 |      70.7 |      81.7 |
> | antmaze-large-diverse-v2 | DGN  |     57.7 |      81.0 |      88.7 |      89.0 |
> |                          | RLPD |     32.7 |      78.0 |      82.0 |      88.0 |
>
> > Was the potential to learn from suboptimal demonstrations considered?
>
> We focus on learning from expert demonstrations, which are not hard to obtain in most practical settings. However, the Antmaze results we provided in the table above shows that DGN can work with non-expert data as well.
>
> >  The policies are trained for a relatively small number of timesteps, and it's unclear if they have fully converged. Furthermore, the plots do not specify the measures of centrality and spread (e.g., mean and standard deviation/error over seeds)
>
> In each plot we run at least three seeds for every plotted series, showing mean and standard error in the mean. We have modified the appendix to specify this. We trained for a number of timesteps sufficient for DGN to converge for a given task.
>
> > Could you clarify how the base policy πθ is used in the action selection process during rollouts?
>
> The base RL policy is a deterministic policy parameterized solely by a policy mean $\mu_\theta$. During rollouts, an action is computed given state s by computing $\mu_\theta(s)$ and then adding noise sampled from $N(0, \Sigma_\phi(s))$. During test-time rollouts, we just use the action  $\mu_\theta(s)$ from the RL policy and do not add sampled noise.
>
> > Figure 2...does not appear to be referenced in the main body of the text.
>
> Thank you for highlighting this. We have added a reference to it in the text.
>
> > could the covariance learning objective be modified to give more weight to action differences from high-reward trajectories?
>
> This is an interesting idea for future research.

---

### Meta-Review · Area_Chair_4sBh · 2026-01-07

**Summary:**

The paper considers the problem of sample-efficient reinforcement learning (RL) in sparse-reward continuous control environments in settings where demonstration data is available. Rather than utlizing the demonstration data via conventional imitation learning (IL), the paper proposes Data-Guided Noise (DGN), a framework that uses demonstrations solely to guide exploration. More specifically, DGN learns the policy mean via standard off-policy RL, while learning a state-dependent exploration noise covariance matrix from the discrepancy between the policy's current action and that of the expert demonstrations. This learned covariance shapes exploratory perturbations during training, without constraining policy optimization. The paper evaluates DGN on various Adroit and Robomimic tasks, demonstrating improvements in sample efficiency and, in the case of harder learning tasks, gains over prior offline-to-online and imitation-augmented RL baselines.  ￼

The paper was evaluated by four reviewers. Several reviewers appreciate the core idea of using demonstration data to guide exploration rather than constrain learning, and find that it is well motivated and clearly presented, as is the paper more generally. At least three reviewers appreciate the breadth of the experimental comparisons as well as the ablation studies. At the same time, the reviewers raise a few key concerns with the paper as originally submitted. These include a perceived lack of a clear theoretical or intuitive justification for why expert–policy action differences should define an effective exploration noise model, viewing the mechanism as empirically motivated but insufficiently explained. Related, some reviewers are concerned about the need for an annealing schedule that serves to trade off between RL and IL losses, which can be seen as another parameter that requires careful tuning (similar to IL-RL los weighting). Additionally, some reviewers comment that the paper does not demonstrate clear empirical benefits of DQN over the baselines, which may limit the method's impact. Meanwhile, at least two reviewers note that the analysis is limited to high-quality expert demonstrations and question how the method would perform if provided with noisy or otherwise suboptimal demonstration data. Additionally, several reviewers comment that the paper omits some common approaches to combining RL and IL.

**Reviewer Concerns:**

The authors addressed many of the reviewers' concerns as part of their rebuttal. The authors provided additional intuition for the covariance-learning mechanism through concrete action-space examples, clarified the action-selection procedure in the algorithm, and expanded discussion of convergence, reporting of statistics, and figure references. In an effort to address concerns about DGN's reliance on high-quality demonstration data, the authors presented additional results showing that DGN remains effective with suboptimal or multimodal low-quality demonstrations and argued that the learned noise naturally broadens rather than collapses in such cases. Considering the concern that the method may be prone to indefinite exploration, the authors argued that annealing or shut-off mechanisms will sufficiently reduce the variance. Additionally, the authors include additional comparisons demonstrating that annealing imitation losses in baseline methods does not recover DGN’s performance, supporting the claim that DGN is easier to tune in practice. Meanwhile, the authors offered to update the paper to more broadly capture work that combines IL and RL, which includes references to papers omitted from the original submission.

**Reviewer Scores:**

The AC suspects that the authors' rebuttal would encourage some of the reviewers to raise their score after having addressed some of their key concerns, but it is unlikely that changes would lead to any reviewer championing the paper.

---

### Decision · Program_Chairs · 2026-01-26

Reject